# The Assessment of the Skin-Whitening and Anti-Wrinkling Effects of *Anemarrhena asphodeloides* Bunge Root Extracts and the Identification of Nyasol in a Developed Cream Product

**Myoung Hee Lee [1],\*** , **Dong IL Jang [2]** and **Jongkeun Choi [3]**

1 Department of Cosmetic Science, Daejeon Health Institute of Technology, 21 Chungjeong-ro, Dong-gu, Daejeon 34504, Republic of Korea

2 Cosmeceutical Science Institute, COTDE Inc., 19-3 Ugakgol-gil, Susin-myeon, Dongnam-gu, Chungcheongnam-do, Cheonan-si 31252, Republic of Korea; daniel@cotde.co.kr

3 Department of Chemical and Biological Engineering, Chungwoon University, 113 Sukgol-ro, Michuhol-gu, Incheon 22100, Republic of Korea; jkchoi@chungwoon.ac.kr

\* Correspondence: leemh@hit.ac.kr

**Abstract:** *Anemarrhena asphodeloides* Bunge (*A. asphodeloides* Bunge) root extract contains nyasol as its main ingredient. Nyasol was extracted and prepared as a cosmetic raw material in 95% ethanol. To identify nyasol as a marker compound qualitative analysis was performed using ultra-performance liquid chromatographyi coupled with electrospray ionization–tandem mass spectrometry. Below a nyasol content of 12 μg/mL, the root extract exhibited negligible cytotoxicity. In this concentration range, melanin production in B16F10 mouse melanoma cells decreased as the concentration of nyasol increased, indicating a skin-whitening effect. In addition, an antiwrinkling effect was confirmed by evaluating the inhibition of MMP-1 protein expression in TNF-α-treated HaCaT cells by either *A. asphodeloides* Bunge root extract (>0.31 μg/mL) or nyasol (>0.25 μg/mL). High-performance liquid chromatography-coupled with a photodiode detector array was used to show that our extract contained 5.06 ± 0.01% nyasol. Furthermore, when this analysis method was applied for the quality control of a cream product containing 2 wt.% of *A. asphodeloides* Bunge root extract, the measured content of nyasol (0.1%) was over 90% of the nominal quantity. Therefore, the product was deemed to be within the required quality standards.

**Keywords:** *Anemarrhena asphodeloides* Bunge root; nyasol; tyrosinase inhibition; MMP-1; anti-aging effect; UPLC-ESI-MS; HPLC-PDA

## 1. Introduction

*Anemarrhena asphodeloides* (*A. asphodeloides*) root extract possesses antipyretic, cardiotonic, diuretic, antibacterial, muco-active, sedative, hypoglycemic, and anticarcinogenic pharmacological properties [1,2]. As the main ingredient of *A. asphodeloides*, the rootstock contains approximately 6% saponins, including steroid saponins, such as timosaponin A-I, A-III, B-II, anemarsaponin B, F-gitonin, smilageninoside, degalactotigonin, and nyasol [3–6]. Of these, timosaponin A-III exhibits anticarcinogenic activity in K562 leukemia cell lines, as well as hypoglycemic activity [7–9]. In addition, *A. asphodeloides* contains polyphenol compounds, such as mangiferin, isomangiferin, and neomangiferin, which are xanthone derivatives [10,11]. The rootstock also contains approximately 0.5% mangiferin (chimonin), which has antidiabetic properties.

*A. asphodeloides* is widely used as herbal medicine in China, Japan, and Korea. In Korea, it is cultivated and processed as a primary raw material for herbal medicine. It is listed as "*Anemarrhena asphodeloides* root extract" (AARE) in the Korean standards for cosmetic ingredients and in the International Cosmetic Ingredient Dictionary and Handbook. *A. asphodeloides* is renowned as a cosmetic raw material (Table 1). In particular, Volufiline™

from the French company Sederma is a popular cosmetic raw material because of its high sarasapogenin content, which has various pharmaceutical applications.

**Table 1.** *Anemarrhena asphodeloides* (*A. asphodeloides*) root extracts sold as cosmetic raw materials.

| Product Name | Manufacturer (Country) | Ingredient |
|---|---|---|
| Premier Anemarrhena 100% Extract | Premier Specialties Inc. (USA) | AARE |
| Volufiline | Sederma SAS (France) | AARE and hydrogenated polyisobutene |
| Gimo Extract | The Garden of Naturalsolution Co. (Korea) | AARE, water, and butylene glycol |
| Chimo Liquid E | Ichimaru Pharcos Co. (Japan) | AARE, water, and alcohol |
| Boumdan | SK Bioland (Korea) | AARE, 6 types of plant extracts, water, and butylene glycol |

In a recent study on the functionality of *A. asphodeloides*, it was found that nyasol, an active ingredient of AARE, inhibited melanin biosynthesis and tyrosinase activity, which led excellent skin-whitening, anti-inflammatory, and anti-atopic effects [12]. In addition, an inhibition of the mushroom tyrosinase activity of an AARE obtained by ethanol extraction was evaluated, and the change in the amount of melanin biosynthesized by treating the cells with the extract was measured [13]. In addition, a clinical trial confirmed the viability of using AARE as a natural skin-whitening cosmetic product [14]. The skin-whitening effect of hot-water-extracted *A. asphodeloides* was mainly due to its inhibitory effect on melanin biosynthesis and tyrosinase activity, which was confirmed by evaluating the expressions of tyrosinase, TRP-2, and MITF using a reverse transcription-polymerase chain reaction (RT-PCR) [15–17].

Skin aging is caused by a significant decrease in collagen, a skin-elasticity protein in the connective tissue that exists in the dermal layer of the skin [18–20]. Extracellular matrix degradation occurs by matrix metalloproteinases (MMPs). In particular, collagen degradation occurs by MMP-1, resulting in wrinkles and fine lines [21,22]. Therefore, the research on antioxidants that can remove free radicals, or extracts that can inhibit MMP-1 activity and promote collagen synthesis, is essential for creating products that can delay skin aging and reduce wrinkles.

In this study, we aimed to prepare *A. asphodeloides* Bunge root extract (AARE) and evaluate its quality using ultra-performance liquid chromatography coupled with electro-spray ionization tandam mass spectrometry (UPLC-ESI-MS) and high-performance liquid chromatograph coupled with a photodiode detector array (HPLC-PDA). We also aimed to validate the safety of AARE in cosmetics through in-vitro investigations, and to confirm its potential as a wrinkle-reducing cosmetic raw material by studying the MMP-1-inhibitory activities of both nyasol and AARE. Additionally, we aimed to confirm the antioxidant and skin-whitening effects of AARE through DPPH free radical scavenging activity tests. In summary, it was concluded that nyasol is the main ingredient in AARE and that it could be used safely in minute amounts in cosmetics for its skin benefits.

## 2. Materials and Methods

### 2.1. Chemicals and Materials

B16F10 mouse melanoma cells (Korean Cell Line Bank(Seoul, Korea) were used as the cell line in the experiments. The cell culture media were 1% antibiotic, 10% fetal bovine serum (FBS), phosphate-buffered saline (PBS), and Dulbecco's Modified Eagle's Medium (DMEM), purchased from Gibco (Carlsbad CA, USA). A non-radioactive cytotoxicity assay kit (Promega, USA) was used to evaluate cytotoxicity. Lipopolysaccharide, 3-(4,5-dimethylthiaxo-2-yl)-2,5-diphenyl tetrazolium bromide (MTT), ascorbic acid, dimethyl sulfoxide (DMSO), L-3,4-dihydroxyphenylalanine, tyrosinase, and arbutin were purchased from Sigma-Aldrich (Burlington, Massachusetts, USA). Nyasol standard reagent was purchased from the Korean Medicine Material Bank (Gyeongsan-si, Gyeongsangbuk-do, Korea) at the National Institute for Korean Medicine Development.

### 2.2. Sample Preparation

AARE solutions containing nyasol were performed as follows: 5 kg of crushed *A. asphodeloides* Bunge root was submerged in 12.5 L of 95% ethanol for 14 days at room temperature, which was then filtered to obtain the ethanol extracts, these extracts were then concentrated under reduced pressure at 65 °C. The resulting concentrate (221 g) was mixed with purified water (2210 g) at a ratio of 10:1 and stirred to remove water-soluble materials. This process was repeated twice. The obtained precipitates were then mixed with 95% ethanol at a ratio of 5:1, stirred, and filtered to obtain ethanol-soluble materials. The ethanol solution was concentrated under reduced pressure at 65 °C until dry, resulting in a yellow powder (recovered amount 20.5 g) containing approximately 5.06% nyasol ($C_{17}H_{16}O_2$, molecular weight: 252.31), which was quantified using a later-discussed method. Figure 1 shows a photograph of both the plant and dried root of *A. asphodeloides* Bunge.

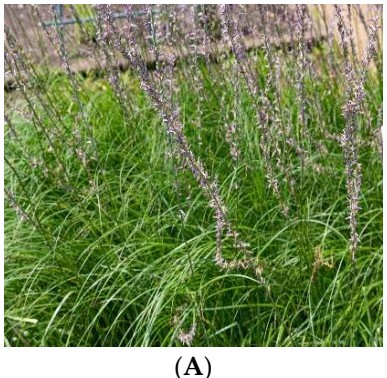 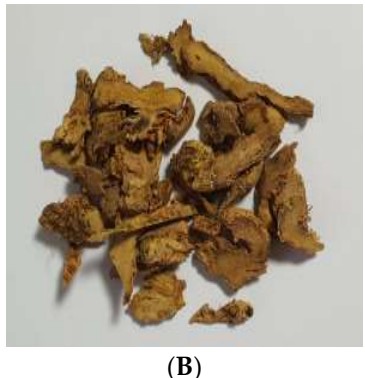

(**A**)  (**B**)

**Figure 1.** Photographs of (**A**) the plant and (**B**) the dried root of *A. asphodeloides* Bunge.

### 2.3. Identification of Nyasol in AAREs Using UPLC-ESI-MS

A nyasol standard (2 mg) was completely dissolved in 10 mL ethanol. The solution was then diluted to nyasol concentrations of 100.00, 50.00, 25.00, 12.50, and 6.250 µg/mL. An AARE sample (50 mg) was dissolved in a sufficient quantity of ethanol to produce 25 mL of solution. An analysis was performed using a Waters Acquity UPLC system, consisting of a liquid chromatography system coupled with an Acquity photodiode detector array (PDA) and a Micromass Quattro Micro API mass spectrometer equipped with an electro-spray ionization (ESI) source. The liquid-chromatography mass- spectrometry parameters were as follows: a cone voltage of 30.0 V, capillary voltage of 3.5 kV, extractor voltage of 3.0 V, source temperature of 120 °C, desolvation temperature of 350 °C, desolvation gas flow of 900 L/h, and cone gas flow of 50 L/h. Mass spectra in the 200–500 m/z range were obtained in the negative-ion mode and the data were analyzed using MassLynx 3.1 software. After injecting 5 µL of the sample solution filtered using a 0.45 µm nylon syringe filter, the sample was separated using a Dionex Acclaim™ 120 C18 column (3 µm, 2.0 mm × 100 mm). The flow rate of the mobile phase was 0.4 mL/min. The mobile phases were 0.1% formic acid in water (A) and 0.1% formic acid in acetonitrile (B). The composition of the mobile phase started at an initial 30% B value and increased linearly to 100% B over 12 min. The experiment was performed for 18 min by changing the solvent composition (Table 2).

**Table 2.** Solvent gradient conditions for UPLC-ESI-MS/MS.

|   | Time (min) | Flow Rate (min) | A (%) | B (%) |
|---|---|---|---|---|
| 1 | 0 | 0.4 | 70 | 30 |
| 2 | 12.0 | 0.4 | 0 | 100 |
| 3 | 13.0 | 0.4 | 0 | 100 |
| 4 | 13.5 | 0.4 | 70 | 30 |
| 5 | 18.0 | 0.4 | 70 | 30 |

After a complete dissolution of the 2 mg nyasol standard in 10-mL of ethanol, the resulting solution was serially diluted to concentrations of 100.00, 50.00, 25.00, 12.50, 6.25, 3.125, 1.562, and 0.7813 μg/mL. AARE (50 mg) was dissolved in ethanol to constitute a 25- mL solution. The prepared standard and AARE solutions were filtered through a 0.45-μm syringe filter, and 10 μL of each solution was injected. An Alliance 2695 (Waters, Milford, MA, USA) separation unit with a 996 PDA system was used for the HPLC analysis of nyasol. A Dionex Acclaim™ 120 C18 column (4.6 mm × 250 mm, 5 μm) was used, and the detector was set at 257 nm. The flow rate was 1.0 mL/min. Distilled water (A) and acetonitrile (B) were used as the mobile phases. The experiment was performed for 20 min by changing the composition of the solvent (Table 3).

**Table 3.** Solvent gradient conditions for HPLC-PDA.

|   | Time (min) | Flow Rate (min) | A (%) | B (%) |
|---|---|---|---|---|
| 1 | 0 | 1 | 50 | 50 |
| 2 | 10 | 1 | 0 | 100 |
| 3 | 13 | 1 | 0 | 100 |
| 4 | 14 | 1 | 50 | 50 |
| 5 | 20 | 1 | 50 | 50 |

### 2.4. Calibration Curve of Nyasol

A nyasol calibration curve was obtained using the prepared standard solutions for concentrations of 100.00, 50.00, 25.00, 12.50, 6.25, 3.125, 1.562, and 0.7813 μg/mL. The calibration curve was plotted with the x-axis as the concentration and the y-axis as the peak area of the nyasol standard. The correlation coefficient ($R^2$) was obtained to evaluate the linearity of the prepared calibration curve. The detection and quantitation limits for each ingredient were determined by signal-to-noise ratios (SNRs) of 3 and 10, respectively.

### 2.5. Quantitative Analysis of Nyasol in the Cream with AARE by HPLC-PDA

A 10 mL solution containing 2 mg of nyasol in ethanol was filtered using a 0.45 μm syringe filter. Then, 1.0 mL of the solution was diluted in ethanol up to 10 mL for use as the sample solution. In addition, 2 mg of nyasol standard was separately dissolved in ethanol to prepare another 10 mL of solution. After extracting 1.0 mL of this solution, a 10-mL solution was prepared with ethanol and used as the standard solution. The peak areas of nyasol in the AARE solutions and nyasol standard were measured by HPLC-PDA using 10 μL of each sample and standard solution.

### 2.6. Statistical Analysis

All values were expressed as the mean ± standard deviation (SD) of the values obtained from the three experiments. The relative standard deviation (RSD, %) was expressed as SD/(mean × 100). Analysis of variance (ANOVA) tests were performed using the *t*-test for statistical significance t between the control and experimental groups. Significance was defined at the $p < 0.05$ level.

### 2.7. Cell Viability Assessment

The experiments were conducted to evaluate the skin-whitening effect of AARE by measuring the total melanin produced in cells with different treatment concentrations of AARE to verify its cytotoxicity. In a previous cytotoxicity experiment [23], $0.5 \times 10^5$ cells/mL of B16F10 melanoma cells per well was placed in a 96-well plate (167008, Nunc™, Waltham, MA, USA, Thermo Fisher Scientific) and cultured in DMEM supplemented with 10% FBS for 24 h. The culture medium was then centrifuged at 3000 rpm for 5 min. To evaluate the cytotoxicity, 50 μL of the MTT solution at a 2 mg/mL concentration was treated and reacted for 4 h to reduce MTT. After the reaction was completed, the culture medium was removed and the microplate was washed with PBS. We added 200 μL of DMSO to the product to completely dissolve the formed formazan crystals. The absorbance was measured at

540 nm using an enzyme-linked immunosorbent assay (ELISA) reader (Biochrom Asys UVM 340 Microplate Reader; Biochrom, UK). After obtaining the average absorbance value for each sample, the cytotoxicity was assessed by comparing the average value with the absorbance value of the control group. The concentrations of AARE used to assess the cell viability of B16F10 melanoma cells were 0, 4, 6, 8, 10, and 12 μg/mL. Arbutin was used as a positive control at concentrations of 10 and 100 μg/mL. The average absorbance value of each sample was obtained, and the cell viability was assessed by comparing the absorbance value of the control group treated with the α-melanocyte-stimulating hormone (α-MSH). All experiments were conducted in triplicate.

An experiment was conducted to determine the concentrations of AARE that are non-toxic to cells, to evaluate the wrinkle-improvement effect through intracellular MMP-1 protein expression determination. A cytotoxicity assessment of human dermal fibroblast (HDF) cells was also conducted. HDF cells were placed in a 96-well plate at $0.1 \times 10^4$ cells/well and incubated for 20 h in a 5% $CO_2$ incubator (Thermo, Korea) at 37 °C. Serial sample dilutions were prepared. The medium from each well in the plate was removed and the wells were washed (200 μL/well, two times) with PBS. The prepared sample dilutions were added to each well containing the HDF cells. After incubating the cells for 24 h in the 5% $CO_2$ incubator at 37 °C, the supernatant from each well was removed. The cells were washed with PBS and 100 μL of HyClone medium was added to each well. After adding 100 μL of MTT solution, the cells were incubated at 37 °C in the 5% $CO_2$ incubator for 4 h. After incubation, the supernatant was removed and the cells were washed with PBS. The precipitate (formazan, a product of MTT reduced by mitochondria) was completely dissolved by adding 200 μL of DMSO. The absorbance was measured at 540 nm using the ELISA reader.

## 2.8. Measurement of Inhibitory Activity of Melanin Biosynthesis

B16F10 melanoma cells were cultured in a 6-well plate for 24 h at $1 \times 10^5$ cells/well and then cultured for 72 h. The medium was replaced with a new DMEM containing 100 nM α-MSH, to which the test samples were added. After incubating the samples for 72 h, the melanin contents were determined. By measuring the absorbance at 405 nm, the concentration of melanin liberated into the medium was evaluated by sampling the supernatant. The concentration of melanin in the cells was obtained by dissolving the cells in 1 M NaOH [24]. The determined melanin contents were normalized by the total protein content. As a positive control for comparison, arbutin (10 and 100 μg/mL) was used. The amount of protein was quantified according to the Bradford method using a bovine serum albumin (BSA) standard solution. After preparing serial dilutions of BSA at concentrations of 0, 2, 5, 10, 20, and 40 μg/mL in distilled water, 20 μL of each BSA dilution was added to 0.98 mL of Bradford reagent. After reacting for 5 min, the solution absorbance was measured at 595 nm to obtain a BSA standard curve. In the same way, the absorbance of the cell lysate of B16F10 melanoma cells cultured in a 6-well plate was measured, and the amount of protein was calculated using the BSA standard curve.

## 2.9. Assessment of MMP-1 Expression-Inhibition Activity

HDF cells at $0.5 \times 10^5$ cells/well were placed in a 48-well plate (150687, Nunc™, Thermo Fisher Scientific) and incubated for 20 h in the 5% $CO_2$ incubator at 37 °C. Serial dilutions of samples were prepared. The media were removed from each well of the plate and the cells were washed (250 μL per well, two times) with PBS. HDF cells in each well were treated with the prepared sample dilutions. After culturing for 24 h in the 5% $CO_2$ incubator at 37 °C, the supernatant of each well was transferred to a 1.5 mL microtube (MCT-150-C, Axygen, Darmstadat, Germany) and centrifuged (12,000 rpm for 5 min) using a Smart-R17 centrifuge (Hanil, Gimpo, Korea). The samples were stored at −20 °C until ELISA was performed.

The Human Pro-MMP-1 ELISA (DMP100, R&D) procedure followed. First, 100 μL of assay diluent RD1-52 was dispensed into each well of an antibody-coated microtiter

plate. Then, 100 µL of the standard and sample was added to each well in which assay diluent RD1-52 was dispensed, and then left at 20–25 °C for 2 h. Then, the supernatant was discarded and the well was washed by dispensing 300 µL of a washing solution (3 times). After dispensing 200 µL of the human pro-MMP-1 conjugate, each well was left at room temperature for 2 h. The supernatant was discarded and each well was washed with 300 µL of washing solution (3 times). After dispensing 200 µL of the substrate solution, each well was left in the dark for 20 min. After adding 50 µL of stop solution, the absorbance was measured at 450 nm with the ELISA reader.

### 2.10. DPPH Radical Scavenging Activity

The antioxidant activity of the prepared AARE was measured by radical scavenging activity using 1,1-diphenyl-2-picrylhydrazyl (DPPH) [25]. Mixed solutions of 0.1-mM DPPH and a sample solution diluted in methanol at various concentrations were prepared and reacted at 37 °C for 30 min to measure the reducing power of each sample against DPPH radicals. The absorbance was measured at 516 nm using a UV-visible spectrophotometer. Ascorbic acid was used as a positive control to compare the activity. The DPPH radical scavenging activity was calculated using Activity (%) = $[1 − (A_E − A_B)/A_C] × 100$, where $A_E$, $A_B$, and $A_C$ are the absorbance values of the sample–DPPH mixture, pure sample solution, and pure DPPH solution, respectively.

## 3. Results and Discussion

### 3.1. Identification of Nyasol in AARE Using UPLC-ESI-MS

A total ion chromatogram of 95% pure nyasol standard solution was obtained by UPLC-ESI-MS analysis. The mass spectrum of the peak at 4.7 min obtained using the negative-ion mode is illustrated in Figure 2A. A molecular ion [M-H]$^−$ at *m/z* 251 was detected in the mass spectrum, corresponding to nyasol. A total ion chromatogram of the prepared AARE was obtained under the same conditions, as shown in Figure 2B. The AARE sample showed a major peak at the same retention time as that obtained from the nyasol standard solution, indicating that nyasol was the main ingredient [11].

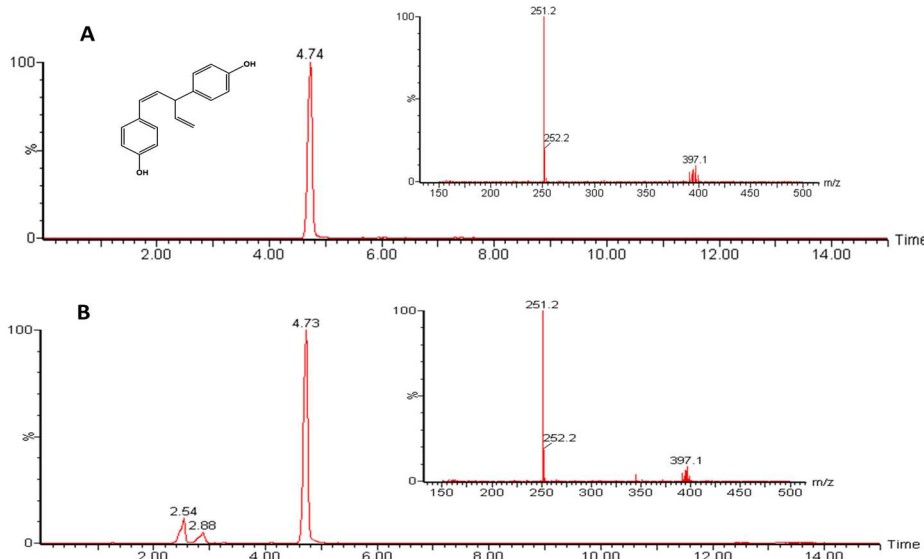

**Figure 2.** Total ion chromatograms of (**A**) nyasol standard solution with a major peak at 4.74 min and (**B**) AARE with a major peak at 4.73 min.

### 3.2. Quantification of Nyasol by HPLC-PDA

For the quality control of AARE an analysis method for nyasol using HPLC-PDA was developed based on UPLC-ESI-MS. The calibration curve obtained from the peak areas of nyasol standard solutions with concentrations of 3.125–100.00 µg/mL is illustrated in

Figure 3. The $R^2$ of the calibration curve was 0.9995, showing excellent linearity. For both SNRs of 3 and 10, the limit of detection (LOD) was 0.05 µg/mL and the limit of quantitation (LOQ) was 0.20 µg/mL. From this calibration curve, we estimated the content of nyasol in the prepared AARE to be 5.06 ± 0.01% (RSD: 0.20%).

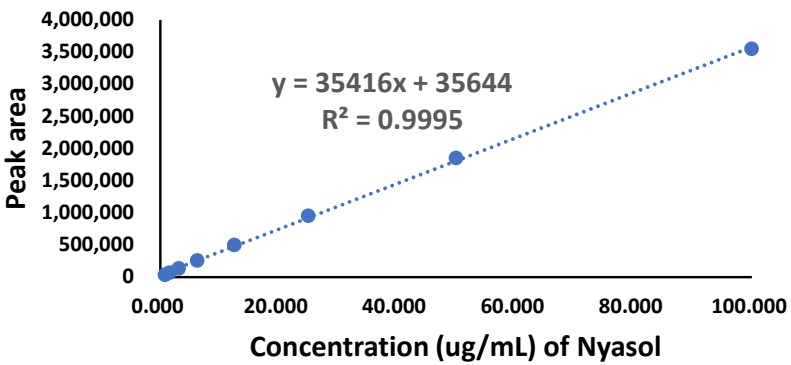

**Figure 3.** Calibration curve of nyasol.

The HPLC-PDA method for nyasol quantification was applied to a developed cream product containing AARE. First, as a result of confirming the specificity of the analysis method, the nyasol peak in the HPLC chromatogram of the sample solution (Figure 4B) did not overlap with other peaks, and the major peak in the UV spectrum (see insets) matched with that of a nyasol standard solution. Therefore, we concluded that the specificity of the analysis was high. The cream used for the analysis contained 2% AARE. Because the nyasol content in our AARE was 5.06 ± 0.01% (RSD: 0.2%), the nyasol content in the cream product corresponded to 0.101%. Based on a quantitative analysis of three different lots of the product, the content of nyasol was in the range of 0.0973–0.0979% (Table 4). Therefore, the product met the quality-control criterium that states that it should contain more than 90% of the nominal nyasol quantity.

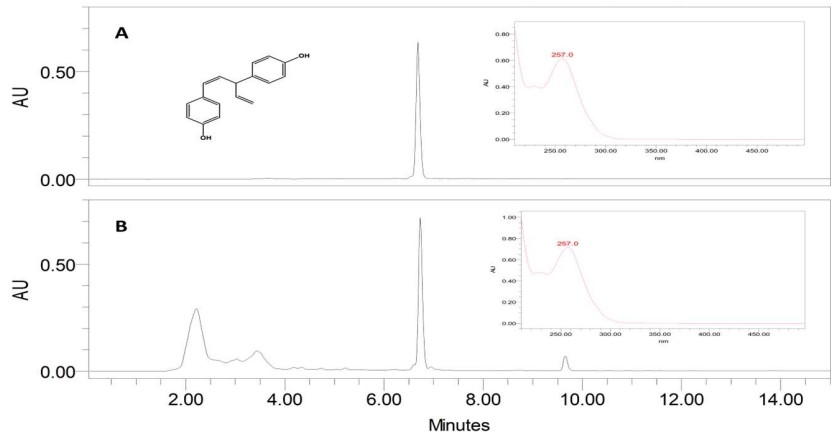

**Figure 4.** HPLC chromatograms of (**A**) nyasol standard solution and a (**B**) cream containing AARE (peak at 6.73 min). Inset: UV spectra.

**Table 4.** Nyasol content (%) in a cream containing 2% AARE and recovery rate (%).

| Sample | Nyasol Content (%) | Recovery Rate (%) | RSD of Recovery Rate (%) |
|---|---|---|---|
| 1 | 0.0974 ± 0.0001 | 96.44 | |
| 2 | 0.0973 ± 0.0006 | 96.34 | 0.33 |
| 3 | 0.0979 ± 0.0006 | 96.93 | |

Recovery rate (%): the analyzed content of nyasol content (g)/the calculated amount of nyasol in 2% AARE cream (g) × 100.

### 3.3. Cell Viability and Melanin Inhibition

The experimental results are expressed as a percentage relative to the control group by setting the cell viability of the control group to 100%. The AARE solutions with concentrations of 4–12 µg/mL exhibited more than 95.0% cell viability (Figure 5). Therefore, all further experiments were performed with AARE concentrations below 12 µg/mL, as it was confirmed that such concentrations had no significant effect on cell viability. In previous studies, nyasol exhibited toxicity at concentrations of 100 µg/mL or higher, whereas at concentrations of 25 µg/mL or less its toxicity was low [11]. Compared with these results, the toxicity of the AARE containing 5% nyasol was low at 12 µg/mL or less. The isolation of nyasol is very difficult and complicated; therefore, the concentration of AARE exhibiting low toxicity similar to nyasol must be defined because pure nyasol is expensive for use as a cosmetic raw material.

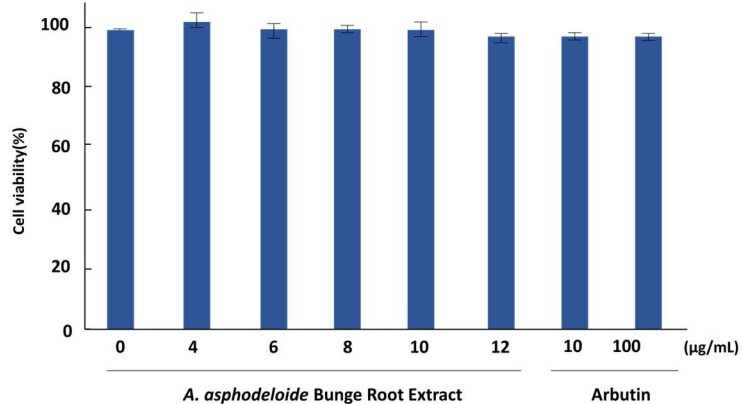

**Figure 5.** Effect of AARE on the growth of B16F10 melanoma cells. Cytotoxicity is measured by MTT assay of AARE and positive control in DMSO-treated B16F10 melanoma cells after 24 h.

To confirm the skin-whitening effect of AARE, the amount of melanin production induced by $\alpha$-MSH in the cells was compared using B16F10 melanoma cells. $\alpha$-MSH promotes melanin synthesis by increasing the activity of tyrosinase in a cell. The cells were treated with 100 nM of $\alpha$-MSH and various concentrations (0, 4, 6, 8, 10, and 12 µg/mL) of AARE, and the amount of melanin was measured after 48 h of incubation. Compared with the test groups treated with arbutin (10 and 100 µg/mL) as a positive control, the AARE inhibited melanin synthesis in a concentration-dependent manner (Figure 6).

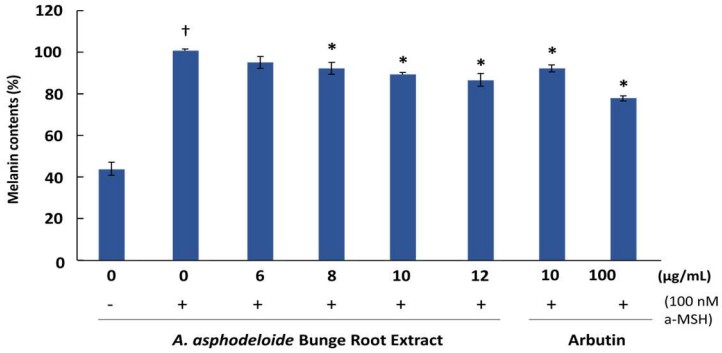

**Figure 6.** Inhibition effect of melanin synthesis in B16F10 melanoma cells by AARE. Changes in melanin contents that were pretreated with 100 nM $\alpha$-MSH and then cultured with AARE for 48 h. Arbutin was used as the positive control. "*": $p < 0.05$; "†": Control without $\alpha$-MSH.

*3.4. Effects of AARE and Nyasol on HDF Cell Viability and TNF-α-induced MMP-1 Protein Expression*

After the treatment of the HDF cells with AARE (0.313, 0.625, 0.780, 1.56, and 2.5 µg/mL), almost 90% of the cells were alive at concentrations of 0.313 and 0.625 µg/mL, indicating low toxicity (Figure 7). In addition, after the treatment of HDF cells with 0.063–0.5 µg/mL of nyasol, more than 90% of the HDF cells were alive, indicating a minor toxicity (Figure 8). These results indicate that the concentration range in which AARE (an economical cosmetic raw material) exhibits low toxicity and wrinkle improvement is higher than when nyasol is used alone.

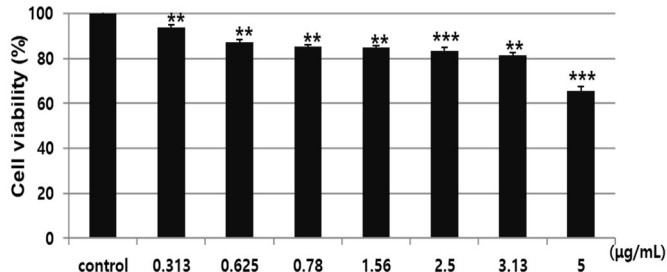

**Figure 7.** Cell viability of HDF cells exposed to various concentrations of AARE determined by an MTT assay ** $p < 0.01$, *** $p < 0.001$ compared to an untreated control.

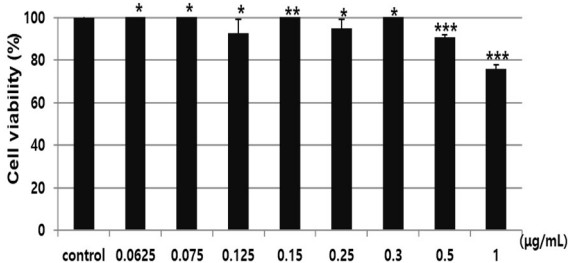

**Figure 8.** Cell viability (MTT) assay of HDF cells treated with nyasol. * $p < 0.05$, ** $p < 0.01$, and *** $p < 0.001$ compared to an untreated control.

Figure 9 illustrates the experimental results of TNF-α-induced MMP-1 protein production, which is a cause of wrinkles, when exposed to various AARE solutions. MMP-1 protein expression was significantly increased after the treatment of the HDF cells with TNF-α (50 ng/mL), whereas subsequent treatment with AARE solutions caused a decrease in the MMP-1 protein expression with increasing AARE concentrations (72.7% expression for 0.625 µg/mL AARE). When 0.0625–0.5 µg/mL nyasol was added to TNF-α-treated HDF cells (50 ng/mL), the expression was suppressed in a concentration-dependent manner (Figure 10). In particular, a significant decrease in the MMP-1 protein expression was observed at concentrations of 0.5 µg/mL (down to 68.3% expression). In addition, the protein expression was significantly decreased in the HDF cells treated with both TNF-α and AARE compared to treatment with TNF-α only.

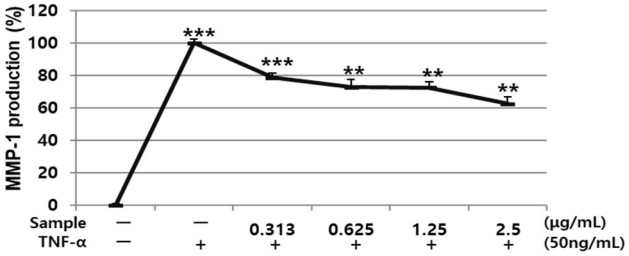

**Figure 9.** Effect of AARE on 50 ng/mL TNF-α-induced MMP-1 production in HDF cells. ** $p < 0.01$, *** $p < 0.001$ when compared to an untreated control.

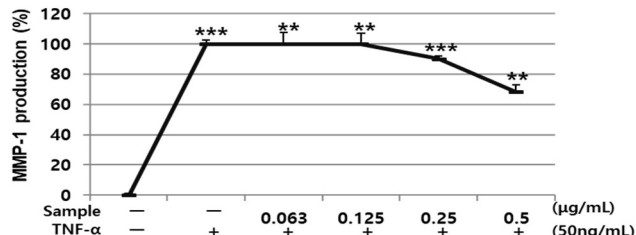

**Figure 10.** Effect of nyasol on 50 ng/mL TNF-α-induced MMP-1 production in HDF cells. ** $p < 0.01$, *** $p < 0.001$ compared to an untreated control.

### 3.5. Antioxidant Effects Using DPPH Free Radical Scavenging Activity Test

In this experiment, the free radical scavenging activity of the AARE was compared to that of ascorbic acid as a control group. The IC$_{50}$ was calculated as the concentration at which 50% of the free radicals were scavenged by the test material. The IC$_{50}$ of the AARE was $157 \pm 10.3$ μg/mL (RSD = 5.29%), which was significantly higher than that of the control group (2.06 μg/mL; RSD = 6.56%), indicating lower activity. Nevertheless, the antioxidant effect was confirmed [11].

### 4. Conclusions

In this study, AARE was prepared as a cosmetic additive by macerating *A. asphodeloides* Bunge root extract using 95% ethanol. UPLC-ESI-MS analyses confirmed that nyasol was the main ingredient in AARE. The analysis of nyasol using HPLC-PDA for the quality control of AAREs produced a linear calibration curve for standard solutions over a wide concentration range, with low LOD and LOQ values. The content of nyasol in the developed AARE was evaluated from the calibration curve. The quantity of nyasol in the product containing 2% AARE was confirmed to be more than 90% of the nominal quantity, with a recovery rate of 96%. The proposed analytical procedure should contribute to better quality control of cosmetic products containing AARE. The in-vitro investigation conducted in this study aimed to validate the safety of AARE in cosmetics, and the findings suggested that the nyasol molecule, which is safe in small quantities, is responsible for the extract's beneficial effects on the skin, making it a valuable addition to the field of cosmeceuticals. The MMP-1 inhibitory activities of nyasol and this extract were studied to confirm the possibility of using AARE as a cosmetic raw material for wrinkle reduction. Both the AARE and nyasol standard inhibited the production of MMP-1, suggesting a wrinkle-reducing effect. In addition to the known skin-whitening effect of AARE, the antioxidant effect of AARE was confirmed by DPPH free radical scavenging activity tests.

**Author Contributions:** M.H.L. led the experiments and interpreted the data in the analysis method and in vitro experiments and composed the manuscript. D.I.J. suggested the experiment idea in the material selection and performed the sample preparation and designed the experiments. J.C. performed the UPLC-ESI-MS and HPLC-PDA analyses. All authors have read and agreed to the published version of the manuscript.

**Funding:** This study was supported by a research program funded by the Daejeon Health Institute of Technology, in 2021.

**Institutional Review Board Statement:** Not applicable.

**Informed Consent Statement:** Not applicable.

**Data Availability Statement:** All data generated or analyzed during this study are included in this published article.

**Conflicts of Interest:** The authors declare no conflict of interest.

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
