# Peer review of "The Assessment of the Skin-Whitening and Anti-Wrinkling Effects of Anemarrhena asphodeloides Bunge Root Extracts and the Identification of Nyasol in a Developed Cream Product"

_cosmetics, doi:10.3390/cosmetics10030073_

Round 1

Reviewer 1 Report

Please be more precise in presenting the aim of your study. I think it is better to present why you are performing these investigations, what are your expectations and what is the novelty, rather than to list what is done in the study.

I think that section material and methods should be improved. It is not correct to present the methodology in a narrative manner. The comment is valid for the whole section. For example: it would be better to present as: A 2,210 g of purified water was added to 221 g of recovered concentrate (1:10 ratio) …

I think that section 2.7., 2.8., 2.9. and 2.10 should be moved forward and be positioned after section 2.2. Sample preparation. The same order should be followed in results and discussions.

The manuscript needs moderate improvement of the language especially in the materials and methods section.

Reviewer 2 Report

This is an in vitro investigation that tries to confirm the safe use in cosmetics of the 2% extract of the Anemarrhena asphodeloides root, a product already used commercially by some cosmetic manufacturers. The work tries to justify that the effects on the skin of this extract are due mostly to the Nyasol molecule, eminently toxic, but harmless in very small concentrations. Therefore it is a study that can be framed within Cosmeceuticals.

I think the work can be published with the following modifications:

Put in the title A. asphodeloides Bunge (Asparagaceae).

Insert a clear photo of the plant and put in section 2.2 the number of the plant control found in a herbarium close to where the plant and its roots were collected.

Solve the chemical structure of Nyasol.

Improve the quality of the figures, for example by putting them in color.
